# Astaxanthin Alleviates Oxidative Stress in Mouse Preantral Follicles and Enhances Follicular Development Through the AMPK Signaling Pathway

**DOI:** 10.3390/ijms26052241

**Published:** 2025-03-02

**Authors:** Jiaqi He, Yue Zhong, Yaqiu Li, Sitong Liu, Xiaoyan Pan

**Affiliations:** Center for Reproductive Medicine, Jilin Medical University, Jilin 132013, China; hjq0000703@163.com (J.H.); zhongyue0227@163.com (Y.Z.); liyaqiu@jlmu.edu.cn (Y.L.); liusitong@jlmu.edu.cn (S.L.)

**Keywords:** preantral follicles, in vitro culture, astaxanthin, mitochondria, oxidative stress

## Abstract

This study investigates the effects of astaxanthin on oxidative stress, mitochondrial function, and follicular development in mouse preantral follicles, with a focus on the involvement of the adenosine monophosphate-activated protein kinase (AMPK) signaling pathway. Astaxanthin (2.5 nM) significantly enhanced both the antrum formation (from 85.96% in the control group to 94.38% in the astaxanthin group) and maturation rates (from 79.15% to 85.12%) of oocytes (*p* < 0.05). From day 4 of in vitro culture, astaxanthin notably increased the area of follicle attachment (from 0.06 µm^2^ to 0.32 µm^2^) and the secretion of estradiol (from 32.10 ng/L to 49.73 ng/L) (*p* < 0.05). Additionally, it significantly decreased malondialdehyde content (from 80.54 μM to 62.65 μM) within the follicles while increasing the mRNA expression levels of glutathione and superoxide dismutase 1 (*p* < 0.05). Astaxanthin also reduced reactive oxygen species levels in oocytes (*p* < 0.05). Notably, astaxanthin enhanced the expression of p-AMPK and PGC-1α, which are key proteins for the AMPK pathway; NRF1 and TFAM, which are crucial for mitochondrial biogenesis; NRF2 and HO-1, which protect against oxidative stress; CO1, CO2, CO3, ATP6, ATP8, and TOM20, which are essential for electron transport chain activity and ATP synthesis; PINK1, Parkin, and LC3-II, which are involved in mitophagy; Bcl-2, which inhibits cell apoptosis; and StAR and P450scc, which promote estrogen synthesis (*p* < 0.05). Furthermore, astaxanthin improved mitochondrial membrane potential and decreased the expression of cleaved caspase 3, Bax, and P53, which promotes cell apoptosis (*p* < 0.05). However, these changes induced by astaxanthin were completely reversed by AMPK inhibitors, indicating the involvement of the AMPK pathway. Conclusively, astaxanthin enhances the in vitro development of follicles, alleviates oxidative stress in preantral follicles, and promotes mitochondrial function during in vitro culture, which may be mediated by the AMPK pathway.

## 1. Introduction

The ovaries in mammals contain a substantial number of preantral follicles. Methods for the in vitro culture of preantral follicles have been established [1,2]. This advancement allows for the application of cultured follicles in evaluating female reproductive health, thereby providing reliable experimental tools for the assessment of new drugs [3]. Moreover, it enhances the availability of oocytes necessary for assisted reproductive technologies and presents a novel approach for fertility preservation in female cancer patients undergoing chemotherapy and radiotherapy [4]. Currently, the predominant method for the in vitro culture of mouse preantral follicles is the two-dimensional droplet technique [5,6,7]. However, the in vitro maturation rate of oocytes remains relatively low, averaging approximately 60% to 70% [5,6,7]. During the in vitro culture of follicles, they are exposed to a hyperoxic environment, which leads to the continuous production of reactive oxygen species (ROS) within the follicles [8]. Excessive ROS can cause the release of cytochrome c from the mitochondria, activating the caspase cascade and resulting in follicular apoptosis [9]. Antioxidants can scavenge ROS, alleviating intracellular lipid peroxidation and DNA fragmentation [10]. Therefore, to alleviate oxidative stress damage, antioxidants have been introduced during the in vitro culture of follicles [11,12,13].

Astaxanthin is a highly potent natural carotenoid with strong antioxidant properties, predominantly found in algae, starfish, crustaceans, salmon, and other marine organisms [10]. It contains hydroxyl and ketone groups on its ionone ring, which contribute to its remarkable antioxidant capacity. The antioxidant activity of astaxanthin is ten times greater than that of other carotenoids (such as canthaxanthin, β-carotene, lutein, and zeaxanthin) and is 100 times stronger than that of α-tocopherol [14]. Astaxanthin has been widely applied in areas such as investigations of anti-tumor, anti-diabetes, anti-inflammation, and anti-obesity due to its antioxidant activity [15,16]. Recently, its applications in the field of reproduction have attracted increasing attention [14,17,18]. Li et al. [19] have reported that astaxanthin could alleviate in vitro embryo damage caused by heat stress or oxidative stress, thereby promoting the in vitro development of bovine somatic cell nuclear transfer embryos. Our previous study also found that astaxanthin can relieve oxidative stress damage caused by bisphenol A and reduce ROS production [20].

Adenosine monophosphate-activated protein kinase (AMPK) is a crucial regulatory molecule in cellular energy metabolism, significantly influencing biological processes such as glycolysis, fatty acid oxidation, autophagy, and mitochondrial biogenesis [21]. When activated, AMPK enhances the expression of downstream antioxidant genes, reduces ROS levels in oocytes, and protects mitochondrial function. However, oxidative stress can impair AMPK activity, resulting in oocyte aging, the arrest of germ vesicle breakdown, and the blockage of meiotic maturation [22]. Astaxanthin can activate AMPK and upregulate the expression of transcriptional coactivators and transcription factors such as peroxisome proliferator-activated receptor gamma coactivator 1α (PGC-1α), thereby inducing mitochondrial remodeling, including increased mitochondrial oxidative phosphorylation and fatty acid metabolism [21]. However, it remains unclear whether astaxanthin and AMPK are involved in regulating mitochondrial function and follicular development during the process of alleviating oxidative stress in follicles.

Therefore, we investigated the role of astaxanthin in alleviating oxidative stress damage during the in vitro development of follicles. Mouse preantral follicles were cultured in vitro in the presence of astaxanthin and AMPK inhibitors. Their effects on follicle development, mitochondrial biogenesis, mitophagy, the antioxidant capacity of follicles, follicle apoptosis, and follicle function were evaluated. The underlying mechanism involving the AMPK pathway was also explored. Our findings may provide evidence for improving the efficacy of follicle development in vitro.

## 2. Results

### 2.1. Astaxanthin Enhances the In Vitro Development of Mouse Preantral Follicles

Preantral follicles were isolated from mice and cultured in vitro for 10 days. As shown in Figure 1A, the follicles began to adhere and grow on day 2 of the in vitro culture. On day 4, the granulosa cells in each group proliferated beyond the basement membrane and started to grow outward. The follicular granulosa cells treated with 2.5 nM astaxanthin demonstrated a faster outward growth rate compared to the other groups, resulting in a notable increase in follicle volume. On day 6, the granulosa cells continued to proliferate, leading to an increase in follicle volume, and there was follicular antrum formation. The antrum formation in the 2.5 nM astaxanthin group occurred earlier than in the other groups, while that in the 25 nM astaxanthin group was delayed compared to the normal group. By day 8, with the increasing follicle volume, there was follicular antrum formation in the majority of follicles. On day 10, the follicular antrum continued to enlarge, and the antral follicles approached maturity.

The survival rate, antrum formation rate, and maturation rate of follicles were evaluated. As presented in Table 1, there were no significant differences in the survival rates of follicles among the groups (*p* > 0.05). However, the antrum formation rate and maturation rate in the 2.5 nM astaxanthin group were significantly higher than those in the control, 0.25 nM astaxanthin, and 25 nM astaxanthin groups (*p* < 0.05). There were no significant differences in the antrum formation rate and maturation rate between the 0.25 nM astaxanthin group and the control group (*p* > 0.05). In contrast, the antrum formation rate and maturation rate in the 25 nM astaxanthin group were significantly lower than those in the control group (*p* < 0.05). Therefore, astaxanthin at the concentration of 2.5 nM enhanced follicle development in vitro and was used in subsequent experiments.

### 2.2. Astaxanthin Increases the Area of Follicle Adhesion and the Secretion of Estradiol

The area of adhesion and the levels of estradiol in the control group, DMSO group, and 2.5 nM astaxanthin group were compared, with the results shown in Figure 1B,C. On day 2, there were no significant differences in the area of follicle adhesion among the groups (Figure 1B) (*p* > 0.05). The area of follicle adhesion in the 2.5 nM astaxanthin group on days 4 (0.06 µm^2^), 6 (0.19 µm^2^), 8 (0.19 µm^2^), and 10 (0.32 µm^2^) was significantly greater than that in the control (0.03 µm^2^, 0.09 µm^2^, 0.09 µm^2^, and 0.29 µm^2^) and DMSO (0.03 µm^2^, 0.09 µm^2^, 0.09 µm^2^, and 0.28 µm^2^) groups (*p* < 0.05) (Figure 1B). Furthermore, the secretion levels of estradiol from the 2.5 nM astaxanthin group on days 4, 6, 8, and 10 (32.10 ng/L, 43.60 ng/L, 48.19 ng/L, and 49.73 ng/L) were significantly higher than those of the control group (22.62 ng/L, 34.51 ng/L, 37.41 ng/L, and 42.05 ng/L) and the DMSO group (22.66 ng/L, 34.91 ng/L, 37.44 ng/L, and 41.52 ng/L) (*p* < 0.05) (Figure 1C). Thus, starting from day 4, 2.5 nM astaxanthin significantly promoted the in vitro growth of follicles and the secretion of estradiol, enhancing the secretion function of the follicles.

### 2.3. Astaxanthin Reduces Lipid Peroxidation and Enhances Antioxidant Capacity in Follicles

The levels of malondialdehyde (MDA), as well as the mRNA expression of the glutathione (GSH) and superoxide dismutase 1 (SOD1) genes in the follicles on day 10, were measured. As shown in Figure 2A, the content of MDA in the follicles of the 2.5 nM astaxanthin group (62.56 μM) was significantly lower than that in the control group (80.54 μM) and the DMSO group (81.90 μM) (*p* < 0.05), indicating that 2.5 nM astaxanthin may reduce lipid peroxidation levels in the follicles. Furthermore, quantitative real-time PCR (qRT-PCR) revealed that the mRNA expression levels of the GSH and SOD1 genes in the follicles of the 2.5 nM astaxanthin group were significantly higher than those in the control group and the DMSO group (*p* < 0.05) (Figure 2B), demonstrating that 2.5 nM astaxanthin could promote the expression of reductant genes and antioxidant enzymes in the follicles.

### 2.4. Astaxanthin Reduces the ROS Levels in Oocytes Likely Through the AMPK Signaling Pathway

The ROS levels in the oocytes on day 11 were detected using ROS staining. The representative staining results are presented in Figure 3A. After staining, the ROS exhibited green fluorescence, with stronger green fluorescence indicating higher ROS content. Statistically, compared with the control group, the mean fluorescence intensity of ROS was significantly lower in the oocytes of the astaxanthin group but significantly higher in the astaxanthin + AMPK inhibitor group (Figure 3B) (*p* < 0.05). However, compared with the astaxanthin group, the astaxanthin + AMPK inhibitor group had significantly higher ROS fluorescence intensity (Figure 3B) (*p* < 0.05). These data showed that ROS levels in oocytes were decreased by astaxanthin. However, this was completely abolished by further treatment with an AMPK inhibitor, implying that astaxanthin may reduce the ROS levels through the AMPK pathway.

### 2.5. Astaxanthin Upregulates the Expression Levels of Mitochondrial Biogenesis and Antioxidant Proteins Possibly Through the AMPK Signaling Pathway

The expression of AMPK pathway proteins p-AMPK and PGC-1α, mitochondrial biogenesis proteins nuclear factor erythroid-2-related factor 1 (NRF1) and mitochondrial transcription factor A (TFAM), and antioxidant proteins nuclear factor erythroid-2-related factor 2 (NRF2) and heme oxygenase-1 (HO-1) in the follicles on day 10 was detected using Western blot. The representative Western blot results are shown in Figure 4A. Notably, the expression levels of p-AMPK, PGC-1α, NRF1, TFAM, NRF2, and HO-1 in the astaxanthin group were significantly higher than those in the control group (Figure 4B) (*p* < 0.05). In contrast, the expression levels of these proteins in the astaxanthin + AMPK inhibitor group were lower than those in both the control group and the astaxanthin group (Figure 4B) (*p* < 0.05). Therefore, the expression levels of mitochondrial biogenesis and antioxidant proteins were upregulated by astaxanthin. This was completely reversed by further treatment with an AMPK inhibitor, suggesting that astaxanthin may enhance the mitochondrial biogenesis and antioxidant capabilities of follicles through the AMPK pathway.

### 2.6. Astaxanthin Enhances the Expression of Mitochondrial Genes in Oocytes Possibly Through the AMPK Signaling Pathway

The expression levels of mitochondrial complex IV subunits cytochrome c oxidase subunit 1 (CO1), cytochrome c oxidase subunit 2 (CO2), and cytochrome c oxidase subunit 3 (CO3); complex V subunits ATP synthase F0 subunit 6 (ATP6) and ATP synthase F0 subunit 8 (ATP8); and the translocase of the outer membrane subunit 20 (TOM20) genes in oocytes on day 11 were detected using qRT-PCR, and the results are shown in Figure 5. Compared with the control group, the mRNA levels of CO1, CO2, CO3, ATP6, ATP8, and TOM20 in the astaxanthin group were significantly increased, while those in the astaxanthin + AMPK inhibitor group were significantly decreased (*p* < 0.05). Interestingly, the effects of astaxanthin on these genes were completely antagonized by the AMPK inhibitor, as indicated by the significantly reduced mRNA levels of these genes in the astaxanthin + AMPK inhibitor group (*p* < 0.05). This result suggests that astaxanthin may enhance mitochondrial gene expression in oocytes through the AMPK pathway.

### 2.7. Astaxanthin May Enhance the Mitophagy of Follicles Through the AMPK Signaling Pathway

Western blot was conducted to detect the expression of mitophagy proteins PTEN-induced kinase 1 (PINK1), Parkin, and LC3-II in follicles on day 10. As shown in Figure 6A,B, the expression levels of PINK1, Parkin, and LC3-II proteins in the astaxanthin group were significantly higher than those in the control group (*p* < 0.05). However, after treatment with an AMPK inhibitor, the expression levels of these proteins in the astaxanthin + AMPK inhibitor group were significantly lower than those in both the control group and the astaxanthin group (*p* < 0.05), indicating the complete reversal of the effects of astaxanthin on mitophagy. Thus, astaxanthin may enhance mitophagy in follicles via the AMPK pathway.

### 2.8. Astaxanthin Increases the Mitochondrial Membrane Potential of Oocytes Potentially Through the AMPK Signaling Pathway

The mitochondrial membrane potential in oocytes on day 11 was assessed using a JC-1 probe. When there is normal mitochondrial membrane potential, JC-1 accumulates in the mitochondrial matrix to form J-aggregates and produces red fluorescence. In contrast, if there is apoptosis, the mitochondrial membrane potential decreases or dissipates, causing JC-1 to exist in the cytoplasm as J-monomers, resulting in green fluorescence. A high ratio of red to green fluorescence intensity indicates a normal mitochondrial membrane potential and that the cells are in a healthy state. As shown in Figure 7A,B, the ratio of red to green fluorescence intensity in the astaxanthin group was significantly higher than that in the control group, while that in the astaxanthin + AMPK inhibitor group was significantly lower (*p* < 0.05). Moreover, this ratio in the astaxanthin + AMPK inhibitor group was significantly reduced compared to that in the astaxanthin group, completely reversing the effects of astaxanthin. These data indicate that astaxanthin may enhance the mitochondrial membrane potential of oocytes via the AMPK pathway.

### 2.9. Astaxanthin Likely Regulates the Expression of Apoptosis Proteins in Follicles Through the AMPK Signaling Pathway

To verify the inhibitory effects of astaxanthin on apoptosis, we detected the expression of apoptosis proteins cleaved caspase 3, Bax, Bcl-2, and P53 in follicles on day 10. The Western blot results revealed that the expression levels of pro-apoptotic proteins cleaved caspase 3, Bax, and P53 in the astaxanthin group were significantly lower than those in the control group, while the expression level of the anti-apoptotic protein Bcl-2 was significantly higher (Figure 8A,B) (*p* < 0.05). In the astaxanthin + AMPK inhibitor group, the effects of astaxanthin on apoptosis protein expression were completely abolished. Specifically, the expression levels of cleaved caspase 3, Bax, and P53 were significantly higher than those in both the control group and the astaxanthin group, whereas the expression level of Bcl-2 was significantly lower (Figure 8A,B) (*p* < 0.05). Hence, astaxanthin may inhibit the expression of pro-apoptotic proteins and promote the expression of anti-apoptotic proteins via the AMPK pathway, thereby exerting inhibitory effects on apoptosis.

### 2.10. Astaxanthin Promotes the Secretion of Estradiol by Follicles Potentially Through the AMPK Signaling Pathway

The estradiol levels in the culture medium of follicles on day 10 were measured using ELISA. As shown in Figure 9A, the astaxanthin group had significantly increased estradiol levels (50.54 ng/L) compared to the control group (40.58 ng/L) (*p* < 0.05). However, the estradiol levels in the astaxanthin + AMPK inhibitor group (21.71 ng/L) were significantly lower than those in both the control group and the astaxanthin group (*p* < 0.05), suggesting the complete abolishment of the effects of astaxanthin on estradiol levels. Therefore, astaxanthin may enhance estradiol secretion by follicles via the AMPK pathway.

To validate the role of astaxanthin in promoting estradiol secretion by follicles, we further detected the expression of estrogen synthesis proteins steroidogenic acute regulatory protein (StAR) and P4350 cholesterol side-chain cleavage enzyme (P450scc). Western blot analysis revealed that the expression levels of StAR and P450scc in follicles on day 10 in the astaxanthin group were significantly higher than those in the control group, whereas their expression levels in the astaxanthin + AMPK inhibitor group were significantly lower than those in both the control group and the astaxanthin group (Figure 9B) (*p* < 0.05). Thus, the AMPK inhibitor completely reversed the effects of astaxanthin on estrogen synthesis proteins. This further confirms the promotive effects of astaxanthin on estradiol secretion by follicles and that this effect is possibly mediated by the AMPK pathway.

## 3. Discussion

High-oxygen environments in vitro can induce oxidative stress damage to follicles [23,24]. The supplementation of antioxidants to the culture medium is crucial for alleviating follicular oxidative stress and improving the developmental rate of follicles in vitro [25]. Our previous research [20] has shown that astaxanthin, as a promising antioxidant, can alleviate BPA-induced oxidative stress damage in follicles. However, it remains unclear whether astaxanthin can mitigate oxidative stress in follicles under high-oxygen conditions and enhance their in vitro development. Here, our study found that 2.5 nM astaxanthin significantly improved the in vitro development of preantral follicles and their secretion of estradiol, reduced their oxidative stress levels, and enhanced the antioxidant capacity of the follicles. Furthermore, astaxanthin reduced the ROS levels in oocytes, promoted mitochondrial biogenesis, enhanced the expression of antioxidant enzymes and mitochondrial genes, boosted mitophagy, decreased follicular apoptosis, and improved follicular secretory function, all of which were closely associated with the AMPK pathway. The protective role of astaxanthin in follicular development may play a critical role in enhancing ovarian function and preserving female reproductive health.

The primary factors affecting the in vitro development of follicles include intracellular oxidative stress induced by the high oxygen levels common in the in vitro culture, which disrupts the redox balance. This disruption leads to the accumulation of excess ROS, resulting in follicle apoptosis and atresia [26]. Astaxanthin, a carotenoid found in aquatic products such as salmon, shrimp, and crabs, serves as an antioxidant that mitigates lipid peroxidation and scavenges excess ROS from cells [14,17,18]. Jang et al. reported that astaxanthin significantly enhanced the survival rate of the bovine oviduct epithelial cells and increased the blastocyst development rate of in vitro fertilization embryos by reducing lipid peroxidation and upregulating the expression of antioxidant and Bcl-2 genes [27]. Similarly, Abdel-Ghani et al. supplemented astaxanthin during the in vitro culture of oocyte–cumulus–granulosa complexes derived from bovine antral follicles, which led to significant increases in both blastocyst development rates and the total cell numbers within the blastocysts [17]. In the present study, we obtained consistent results. The introduction of astaxanthin into the in vitro culture medium of mouse preantral follicles resulted in significant decreases in MDA content and substantial increases in the expression of the GSH and SOD1 genes. These changes indicate that astaxanthin could reduce lipid peroxidation and enhance antioxidant capacity, thereby improving the development of mouse preantral follicles in vitro.

Recent studies [28,29] have demonstrated that astaxanthin functions as an activator of AMPK, a crucial energy sensor and metabolic regulator [30]. In response to oxidative stress, astaxanthin facilitates the phosphorylation of the α subunit at Thr172 by AMPK. Activated AMPK further stimulates PGC-1α, which translocates to the nucleus and activates both NRF1 and NRF2. This cascade promotes the transcription of nuclear components encoding the respiratory chain and TFAM, thereby enhancing mitochondrial protein synthesis and promoting mitochondrial biogenesis [31,32], as well as regulating cellular metabolic states. Additionally, the induction of NRF2 by PGC-1α modulates the oxidative stress response [33]. NRF2 is among the most critical transcription factors for regulating cellular oxidative stress and maintaining redox homeostasis. It mediates the expression of various antioxidant proteins involved in defense systems, including quinone oxidoreductase 1 and HO-1, thereby alleviating cellular damage induced by reactive oxygen species and electrophiles [34]. Astaxanthin boosts NRF2 expression in skeletal muscle cells through the PGC-1α pathway, consequently enhancing the expression of antioxidant enzymes glutamate cysteine ligase and NADH Dehydrogenase, quinone 1 on cellular membranes, and mitochondria, which helps to mitigate oxidative stress damage in these cells [35,36]. Remarkably, astaxanthin significantly increases the expression levels of p-AMPK, PGC-1, NRF1, and TFAM in skeletal muscle cells, promoting mitochondrial biogenesis after oxidative stress damage [29]. Consistently, this study found that astaxanthin induced a significant reduction in ROS levels in follicles. Moreover, astaxanthin elevated the expression of p-AMPK, PGC-1α, NRF1, TFAM, NRF2, and HO-1. Additionally, treatment with the AMPK inhibitor completely abolished these effects of astaxanthin. These results suggest that astaxanthin may promote mitochondrial biogenesis, enhance antioxidant capacity, and facilitate the removal of excess ROS in follicles by activating AMPK and PGC-1.

AMPK plays a pivotal role in maintaining mitochondrial homeostasis [37]. Following oxidative stress, the accumulation of ROS in cells leads to the damage of mitochondria, a decrease in mitochondrial membrane potential, and reduced ATP production, subsequently activating AMPK. The activated AMPK then stimulates the PINK–Parkin mitophagy pathway [38]. Normally, PINK1 is synthesized in the cytoplasm and, through the actions of the mitochondrial transport protein complexes TOM and translocase of the inner mitochondrial membrane, is localized to the inner mitochondrial membrane. After this, the N-terminal of PINK1 is degraded by the protease presenilin-associated rhomboid-like protein, while the C-terminal is translocated out of the mitochondria for degradation [39]. Since PINK1 can undergo degradation within mitochondria, its expression level remains low in healthy cells. However, when mitochondria are damaged, the mitochondrial membrane undergoes depolarization, preventing the integration of PINK1 into the inner mitochondrial membrane for degradation, thereby leading to its accumulation on the outer mitochondrial membrane [40]. The accumulation of active PINK1, which possesses kinase activity, recruits and activates Parkin [41]. Upon activation, Parkin ubiquitinates numerous substrates on the outer mitochondrial membrane; these ubiquitinated mitochondria are subsequently connected to LC3-II-positive autophagosomes via autophagy receptors, engulfed by the autophagosomes, and then fused with lysosomes for the degradation of damaged mitochondria [42]. This process is essential for maintaining mitochondrial homeostasis and overall cell health. Once autophagy is initiated, LC3-I is lipidated to form the anchored protein LC3-II, an important marker for assessing autophagosome content [43]. Astaxanthin has been shown to regulate the AMPK/mammalian target of rapamycin/unc-51 like kinase 1 autophagy pathway, enhancing the expression of LC3-II protein and improving skeletal muscle injury in rats [44]. Furthermore, astaxanthin can elevate PINK and Parkin expression in damaged vascular smooth muscle cells, promoting the clearance of damaged mitochondria through mitophagy, thereby maintaining normal mitochondrial homeostasis [45]. Here, our results also demonstrated that astaxanthin increased the expression of PINK1, Parkin, and LC3-II in follicles cultured in vitro, suggesting that astaxanthin may promote the autophagy of damaged mitochondria following oxidative stress. Notably, this effect can be completely reversed by an AMPK inhibitor, implying that astaxanthin may regulate mitophagy through AMPK. Although astaxanthin is crucial for the homeostasis of follicular mitochondria, it remains uncertain whether it also increases the quantity of healthy mitochondria.

The simultaneous upregulation of mitochondrial biogenesis and mitophagy proteins by astaxanthin may initially seem contradictory, as these processes are often considered antagonistic. However, they are complementary mechanisms that work in concert to maintain mitochondrial quality and cellular homeostasis. Mitochondrial biogenesis generates new, healthy mitochondria to replace damaged ones, while mitophagy selectively removes dysfunctional or damaged mitochondria to prevent the accumulation of ROS and maintain a healthy mitochondrial network [46]. Under conditions of oxidative stress, such as those induced by high oxygen levels during in vitro follicle culture, both processes are essential for restoring mitochondrial function and ensuring cellular survival [47,48]. In our study, the upregulation of astaxanthin on both mitochondrial biogenesis (via p-AMPK, PGC-1α, NRF1, and TFAM) and mitophagy (via PINK1, Parkin, and LC3-II) suggests that it promotes a dynamic balance between mitochondrial renewal and degradation. We suppose that this dual action ensures that damaged mitochondria are efficiently removed while new, functional mitochondria are generated to meet the energy demands of the cell. Such a coordinated response is critical for alleviating oxidative stress and improving follicular development.

Previous studies [49,50] have shown that astaxanthin can protect neuronal mitochondrial structures through the AMPK pathway by enhancing the expression of TOM20 and Mitofusin2, thereby maintaining normal mitochondrial quantity. To further confirm the effects of astaxanthin on mitochondrial structures in follicles, we analyzed the expression of genes encoding mitochondrial structural proteins, including complex IV subunits CO1, CO2, and CO3; complex V subunits ATP6 and ATP8; and TOM20. The results demonstrated that the expression of these genes significantly increased after astaxanthin treatment. Interestingly, this effect was reversed completely by the AMPK inhibitor. This suggests that astaxanthin may enhance the expression of mitochondrial structural genes via the AMPK pathway. It has been indicated that mitochondrial biogenesis and function depend on the TOM complex, with TOM20 serving as the principal receptor subunit, responsible for the initial step in mitochondrial protein import [51]. Mitochondrial complexes IV and V are essential components of the electron transport chain, participating in mitochondrial oxidative phosphorylation and ATP production [52]. Mitochondrial complex IV comprises three conserved subunits (CO1, CO2, and CO3) and a variable number of accessory subunits, depending on the organism [53]. The activity of mitochondrial complex IV is regulated by glutathionylation, and pharmacological interventions targeting glutathionylation increase free cysteine levels, thereby enhancing the activity of mitochondrial complex IV and promoting cellular ATP production [54]. ATP6 and ATP8 are two subunits encoded by mitochondrial DNA that comprise mitochondrial complex V [55]. ATP6 is crucial for the coupling mechanism of proton translocation to ATP synthesis, functioning through the rotary catalysis of ATP synthase and ATP8 within the quaternary structure of the enzyme complex [55]. Congenital mutations in the mitochondrial ATP6 and ATP8 genes are recognized as pathogenic, impairing normal energy production in cells and clinically manifesting as a range of neurodegenerative and multisystem diseases [56]. Therefore, the integrity of the structures of mitochondrial complexes IV and V is essential for maintaining proper mitochondrial function. Here, the observed increase in the expression of these genes indicates an elevation in the number of healthy mitochondria, thereby enhancing mitochondrial function.

The accumulation of ROS due to oxidative stress can activate intracellular caspase signaling pathways, leading to apoptosis [45]. Astaxanthin can not only scavenge ROS and reduce MDA levels but also activate the AMPK signaling pathway, thereby enhancing the expression of antioxidant genes such as HO-1, which improves the antioxidant and anti-apoptotic capacities of cells [57,58]. Furthermore, astaxanthin can reduce the expression of cleaved caspase-3 and Bax proteins while increasing Bcl-2 protein expression through the AMPK-SIRT1 pathway, providing protective effects against sevoflurane-induced neuronal apoptosis and oxidative stress damage [20]. In our study, we observed that astaxanthin elevated the mitochondrial membrane potential in oocytes and increased the expression of the anti-apoptotic gene Bcl-2 while decreasing the expression of pro-apoptotic genes Bax and P53, thus preventing the cleavage of caspase-3 and inhibiting apoptosis. P53 serves as a tumor suppressor, plays a critical role in both the cell cycle and apoptosis, and is a downstream factor of AMPK [59]. The anti-apoptotic protein Bcl-2 and the pro-apoptotic molecule Bax are both members of the Bcl-2 family. Oxidative stress and the excessive accumulation of ROS within cells activate Bax protein, which inserts into the outer mitochondrial membrane to form Bax/Bax homodimer channels. This insertion increases mitochondrial outer membrane permeability and facilitates the release of cytochrome c from the mitochondrial intermembrane space into the cytoplasm. Cytochrome c then binds to the adaptor protein apoptotic protease activating factor-1 to form apoptosomes, which subsequently cleave and activate downstream caspases, particularly cleaved caspase-3, thereby inducing apoptosis [60]. Bcl-2 can form heterodimers with Bax, inhibiting Bax translocation and dimerization, effectively suppressing apoptosis [60]. Studies have shown that AMPK can inhibit c-Jun N-terminal kinase phosphorylation [61], thereby reducing Bcl-2 phosphorylation and its role in promoting cytochrome c release, ultimately suppressing the activation of caspase-3 and apoptosis [62]. These findings suggest that AMPK is a significant target for regulating apoptosis under conditions of oxidative stress. Furthermore, this study demonstrates that the inhibitory effect of astaxanthin on follicular apoptosis is notably diminished following AMPK inhibition, indicating that AMPK is a critical pathway through which astaxanthin exerts its anti-apoptotic effects.

A critical function of ovarian follicles is the secretion of estrogen, which serves as a key indicator of follicular health [63]. StAR and P450scc are essential factors in the synthesis of estradiol, located on the outer and inner mitochondrial membranes, respectively. The initiation and rate-limiting step of estradiol biosynthesis is the transfer of cholesterol into the mitochondria, which is mediated by StAR [64]. A decrease in mitochondrial membrane potential impedes StAR-mediated cholesterol transport into the mitochondria [65]. Under the influence of P450scc, cholesterol is converted into pregnenolone via hydroxylation and side-chain cleavage. Pregnenolone is subsequently converted into androstenedione through hydroxylation, cleavage, and dehydrogenation, ultimately transforming into estradiol through estrone. Oxidative stress compromises mitochondrial membrane integrity, resulting in a decreased mitochondrial membrane potential, which negatively impacts StAR-mediated cholesterol transport and reduces estradiol synthesis [66,67,68,69]. Concurrently, a reduction in the synthesis of StAR and P450scc proteins significantly diminishes estradiol synthesis [70]. In this study, we found that astaxanthin enhanced the expression of StAR and P450scc in the follicles, thereby promoting both the synthesis and secretion of estradiol. On the contrary, after AMPK inhibition, the expression levels of StAR and P450scc, along with their associated synthesis and secretion of estradiol in the follicles, were diminished. This phenomenon may be closely related to the role of AMPK in regulating mitochondrial biogenesis and maintaining mitochondrial homeostasis, ultimately increasing the population of healthy mitochondria.

Astaxanthin has shown promising applications in human-assisted reproductive technology. For instance, in women with endometriosis-related infertility, astaxanthin supplementation (6 mg/day for 12 weeks) significantly increased the number of retrieved oocytes and improved embryo quality [71]. Similarly, in patients with polycystic ovary syndrome, astaxanthin treatment (12 mg/day for 60 days or 8 mg/day for 40 days) enhanced the rates of high-quality oocytes and embryos, as well as oocyte maturation rates [72,73,74]. Furthermore, in poor ovarian responders, astaxanthin supplementation (12 mg/day for 8 weeks) improved the number of retrieved oocytes, mature oocytes, frozen embryos, and high-quality embryos [75]. These studies demonstrate that astaxanthin plays a significant role in improving oocyte and embryo quality in humans. While astaxanthin has been widely used in the in vitro culture of oocytes from mice, pigs, and cattle [17,20,76,77], its application in the in vitro culture of human oocytes or follicles remains unexplored, likely due to the challenges associated with obtaining human oocytes and the immaturity of in vitro maturation systems. Nonetheless, these findings underscore the potential of astaxanthin as a therapeutic agent in human-assisted reproductive technology and provide a strong rationale for further investigation into its mechanisms and applications.

## 4. Materials and Methods

### 4.1. Study Animals

Female SPF-grade Kunming mice (14 days old) were purchased from Changchun Yisi Laboratory Animal Center (Changchun, China). The mice were housed in an environment with a temperature of (22 ± 2) °C, a humidity of (50 ± 10)%, and a light cycle of 14 h of light and 10 h of darkness. They had free access to food and water. The animal study protocol was approved by the Ethics Committee of Jilin Medical University (protocol code 2019-KJT011 and 15-11-2022)

### 4.2. In Vitro Culture of Preantral Follicles

Mice (*n* = 80) were euthanized using cervical dislocation, and their bilateral ovaries were quickly retrieved and placed in an L-15 working solution containing 10% fetal bovine serum (#11415064; Gibco, Grand Land, NY, USA). Under a stereomicroscope, surrounding tissues were carefully removed, and the follicles were isolated using a fine separating needle. According to the follicle grading criteria by Pedersen et al. [78], preantral follicles with diameters of 110–130 µm and containing three layers of granulosa cells were selected. They were placed in α-MEM (#12571063, Gibco) culture droplets containing 100 mIU/mL recombinant FSH (r-FSH) (Gonal-F, Serono, Geneva, Switzerland), 5% fetal bovine serum (#10091155, Gibco), and 1% Insulin–Transferrin–Selenium (#41400045, Gibco).

For drug treatment, follicles were cultured in the presence of 0.1% DMSO (#D2650, Sigma, St. Louis, MO, USA); or 0.25 nM, 2.5 nM, or 25 nM of astaxanthin (dissolved in DMSO; #SML0982, Sigma); or 2.5 nM of astaxanthin and 1 μM of AMPK inhibitor (Compound C; dissolved in DMSO; #P5499, Sigma) [22], designated as the DMSO group, 0.25 nM astaxanthin group, 2.5 nM astaxanthin group, 25 nM astaxanthin group, and astaxanthin + AMPK inhibitor group, respectively. In each group, 30 follicles were cultured each time. These cultures were maintained at 37 °C with 5% CO2 for 10 days, with half-quantitative medium changes every two days, while observing and documenting the growth of follicles in each group. A follicle was considered viable if it was attached to the wall on day 2, and a follicle was classified as antral if a follicular cavity appeared on day 8. The survival and antrum formation rates of the follicles were recorded. On day 10, 2.5 U/mL of human chorionic gonadotropin (Lizhu Pharmaceutical Group, Zhuhai, China) was added to the follicle culture medium, and the culture continued for 16 h. Granulosa cells surrounding the cumulus–oocyte complexes were removed using a fine glass pipette. Oocytes that had the first polar body were considered mature oocytes, and the oocyte maturation rate was calculated. The attached area of follicles on days 4, 6, 8, and 10 was analyzed using Image J software, with 10 follicles analyzed per group.

### 4.3. ELISA

On days 4, 6, 8, and 10 of the follicle culture, the follicle culture media were collected from 30 follicles each time. The levels of estradiol in the culture media were detected using an ELISA kit (#BPE20376, Shanghai Lengton Biological Technology Co., Ltd., Shanghai, China).

### 4.4. Detection of MDA Levels in Follicles

On day 10 of the culture, 20 follicles were collected and treated with cell lysis buffer (#P0013, Beyotime, Beijing, China). Lysates were centrifuged at 10,000× *g*–12,000× *g* for 10 min, and the supernatant was collected for further analysis. The MDA levels were measured using an MDA detection kit (#S0131S; Beyotime). The absorbance was measured at 532 nm using a SpectraMax Absorbance Reader (Molecular Devices, Sunnyvale, CA, USA). The MDA content in the sample was calculated based on the standard curve.

### 4.5. qRT-PCR

On day 10, 20 follicles were collected from each group. RNAs were extracted from these follicles using an RNeasy Micro Kit (Qiagen, Hilden, Germany) and then reverse-transcribed into cDNA. The qRT-PCR was conducted on an iQ5 Multicolor Real-time PCR Detection System (Bio-RAD, Hercules, CA, USA). The primer sequences are listed in Table 2. The PCR conditions were as follows: pre-denaturation at 95 °C for 30 s and 40 cycles of denaturation at 95 °C for 5 s, annealing at 59 °C for 20 s, and extension at 72 °C for 30 s. The housekeeping gene β-actin was used as an internal control, and the relative expression levels of target genes were calculated using the 2^−△△Ct^ method.

### 4.6. ROS Level Detection in Oocytes

After treatment with human chorionic gonadotropin for 16 h, cumulus–oocyte complexes were collected. After the removal of cumulus cells, the obtained oocytes were incubated in an L-15 medium containing 10 mM carboxy-H2DCF diacetate (#S0033, Beyotime) at 37 °C for 30 min. After washing three times with L-15 medium, the oocytes were observed under an Olympus IX-83 microscope (Olympus, Tokyo, Japan) and photographed using the cellSens imaging software (version 3.1.1). All images were captured under the same settings. The fluorescence intensity of each oocyte was analyzed using Image J software, with 10 oocytes analyzed per group.

### 4.7. Western Blot

Granulosa cells were collected from 40 follicles cultured in vitro for 8 or 10 days in each group, and proteins were extracted using RIPA lysis buffer (containing 1% PMSF) (#P0013B and #P1006, Beyotime). The proteins were transferred to a PVDF membrane, and the membrane was blocked with 5% non-fat milk for 1 h. After blocking, the membrane was incubated overnight at 4 °C with primary antibodies against p-AMPK (#50081, Cell Signaling Technology, Boston, MA, USA), PGC-1α (#ab191838, Abcam, Cambridge, UK), NRF1 (#46743, Cell Signaling Technology), TFAM (#ab252432, Abcam), NRF2 (#ab92946, Abcam), HO-1 (#ab52947, Abcam), PINK1 (#ab23707, Abcam), Parkin (#32833, Cell Signaling Technology), LC3-II (#43566, Cell Signaling Technology), cleaved caspase 3 (#9664, Cell Signaling Technology), Bax (#2772, Cell Signaling Technology), Bcl-2 (#3498, Cell Signaling Technology), P53 (#ab26, Abcam), StAR (#ab203193, Abcam), P450scc (#ab272494, Abcam), and β-actin (#ab8226, Abcam). After washing, the membrane was incubated at room temperature for 2 h with goat anti-rabbit HRP-conjugated secondary antibody (31210, Thermo Scientific Pierce, Waltham, MA, USA), followed by enhanced chemiluminescence detection. The membrane was scanned using ChemiDOC XRS+ imaging systems (Bio-Rad). The relative expression levels of the target proteins were analyzed using Image J software.

### 4.8. Mitochondrial Membrane Potential Analysis Using JC-1

Oocytes were washed three times with L-15 medium and then incubated in a culture medium containing 0.5 µmol/L JC-1 (Invitrogen, Grand Island, NY, USA) at 37 °C with 5% CO2 for 30 min. Fluorescence images were captured using the CellSense imaging software (version 3.1.1) within the Olympus IX-83 fluorescence microscope. The mitochondrial membrane potential was indicated by the ratio of red to green fluorescence intensity. Ten oocytes were analyzed for each group.

### 4.9. Statistical Analysis

Statistical analysis was performed using SPSS version 17.0. All experiments were repeated three times. The data are expressed as the mean ± standard deviation. Differences among multiple groups were compared using a one-way ANOVA followed by an LSD post hoc test. A *p*-value of less than 0.05 was considered statistically significant.

## 5. Conclusions

This study demonstrates that astaxanthin acts as a potent antioxidant that significantly enhances the in vitro development of mouse preantral follicles by alleviating oxidative stress and promoting mitochondrial function. Specifically, astaxanthin upregulated essential proteins involved in mitochondrial biogenesis, mitophagy, and antioxidant responses while concurrently inhibiting apoptosis. Moreover, astaxanthin not only mitigated ROS accumulation but also improved the overall functional capacity of ovarian follicles by stimulating estradiol secretion. Mechanically, these effects may be achieved through the activation of the AMPK pathway. Our findings underscore the therapeutic potential of astaxanthin as a candidate for enhancing reproductive health, particularly in the context of assisted reproductive technologies where oxidative stress poses a considerable challenge. Future research is warranted to further elucidate the precise molecular mechanisms through which astaxanthin modulates follicular dynamics and to explore its efficacy in clinical applications for fertility preservation and enhancement.

## Figures and Tables

**Figure 1 ijms-26-02241-f001:**
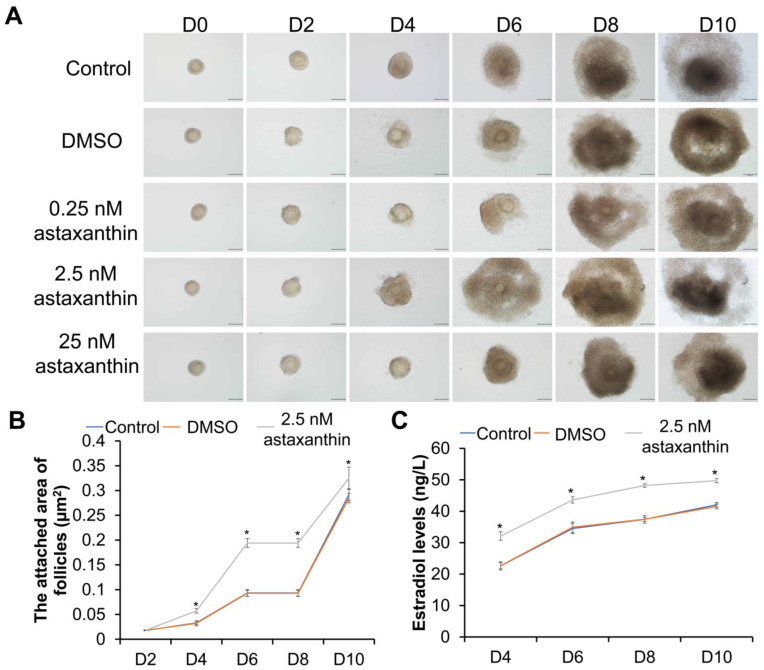
Effects of astaxanthin on in vitro development of mouse preantral follicles. (**A**) In vitro development process of preantral follicles in presence of various concentrations of astaxanthin (scale bar = 100 μm). (**B**) Attached area of follicles from day 2 to day 10 (µm^2^). In each group, 8 follicles were used. (**C**) Secretion of estradiol by follicles from day 4 to day 10 (ng/L). In each group, 30 follicles were used. Experiments were repeated three times independently (N = 3). * *p* < 0.05, compared with control group.

**Figure 2 ijms-26-02241-f002:**
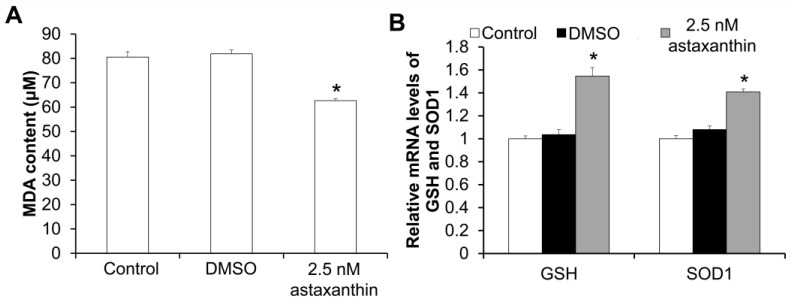
Effects of astaxanthin on MDA content and mRNA expression of GSH and SOD1 in follicles. (**A**) MDA content in follicles (μM). (**B**) qRT-PCR detection of mRNA levels of GSH and SOD1 in follicles. In each group, 20 follicles were used. Each experiment was performed in triplicate (N = 3). * *p* < 0.05, compared with control group.

**Figure 3 ijms-26-02241-f003:**
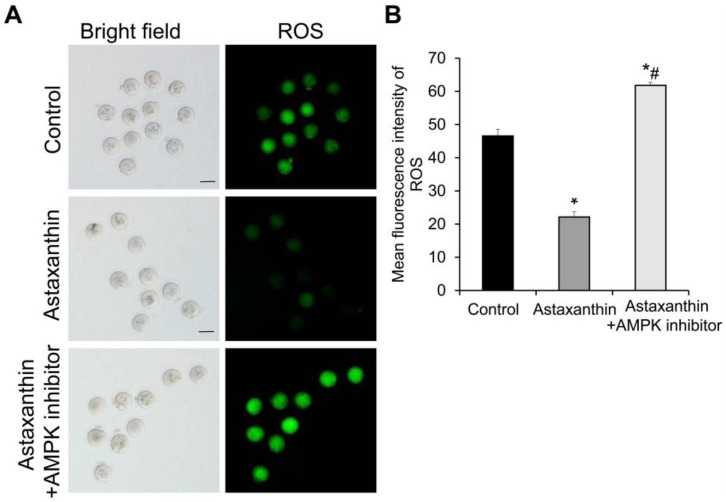
Effects of astaxanthin on ROS levels in oocytes. (**A**) Detection of ROS levels in oocytes using carboxy-H2DCF diacetate. Representative bright-field and fluorescent images are presented. Scale bar: 100 μm. (**B**) Measurement of mean fluorescence intensity of ROS in oocytes using Image J (version 1.8.0) image analysis software. Experiment was performed in triplicate (N = 3), and 8 follicles were used in each group. * *p* < 0.05, compared with control group. # *p* < 0.05, compared with astaxanthin group.

**Figure 4 ijms-26-02241-f004:**
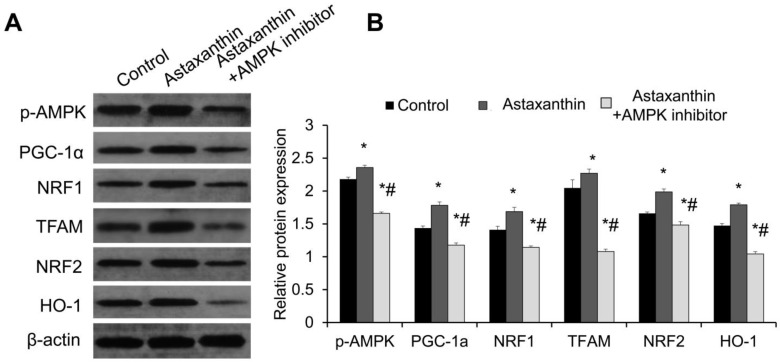
Effects of astaxanthin on expression of AMPK pathway proteins, mitochondrial biogenesis proteins, and antioxidant proteins. (**A**) Western blot analysis of expression of AMPK pathway proteins (p-AMPK and PGC-1α), mitochondrial biogenesis proteins (NRF1 and TFAM), and antioxidant proteins (NRF2 and HO-1) in follicles on day 10. (**B**) Analysis of relative expression levels of these proteins using Image J image analysis software. Experiment was repeated three times independently (N = 3), and 40 follicles were used in each group. * *p* < 0.05, compared with control group. # *p* < 0.05, compared with astaxanthin group.

**Figure 5 ijms-26-02241-f005:**
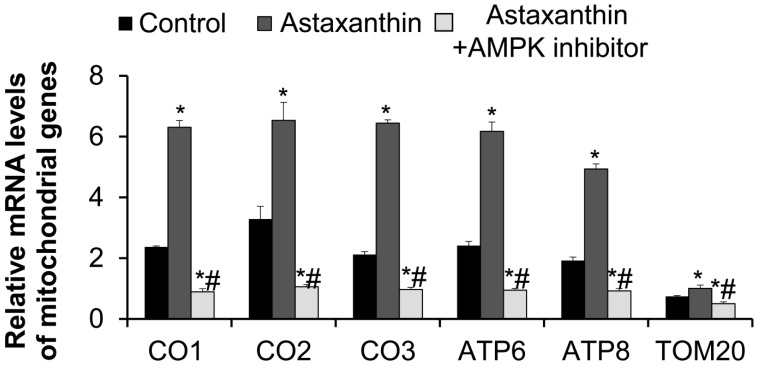
The effects of astaxanthin on the mRNA levels of mitochondrial genes in oocytes. The mRNA levels of mitochondrial genes were detected with qRT-PCR. In each group, 20 follicles were used. The experiment was conducted in triplicate (N = 3). * *p* < 0.05, compared with the control group. # *p* < 0.05, compared with the astaxanthin group.

**Figure 6 ijms-26-02241-f006:**
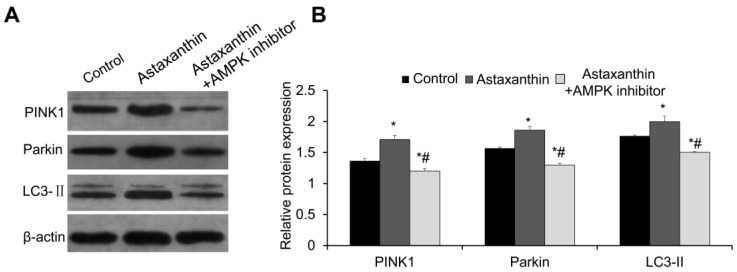
Effects of astaxanthin on expression of mitophagy proteins. (**A**) Western blot detection of expression of mitophagy proteins PINK1, Parkin, and LC3-II in follicles on day 10. (**B**) Analysis of relative expression levels of these proteins using Image J image analysis software. Experiment was performed in triplicate (N = 3), and 40 follicles were used in each group. * *p* < 0.05, compared with control group. # *p* < 0.05, compared with astaxanthin group.

**Figure 7 ijms-26-02241-f007:**
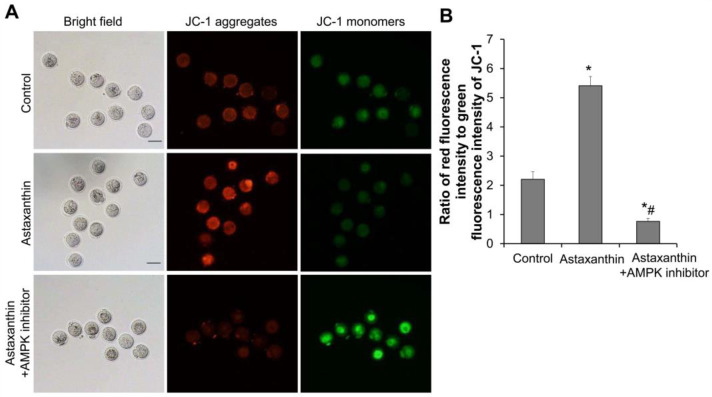
Effects of astaxanthin on mitochondrial membrane potential of oocytes. (**A**) Detection of mitochondrial membrane potential in oocytes using JC-1 probe. Representative bright-field and fluorescent images are shown. Scale bar: 100 μm. (**B**) Ratio of red fluorescence intensity to green fluorescence intensity of JC-1. In each group, 8 follicles were used. Experiment was repeated three times independently (N = 3). * *p* < 0.05, compared with control group. # *p* < 0.05, compared with astaxanthin group.

**Figure 8 ijms-26-02241-f008:**
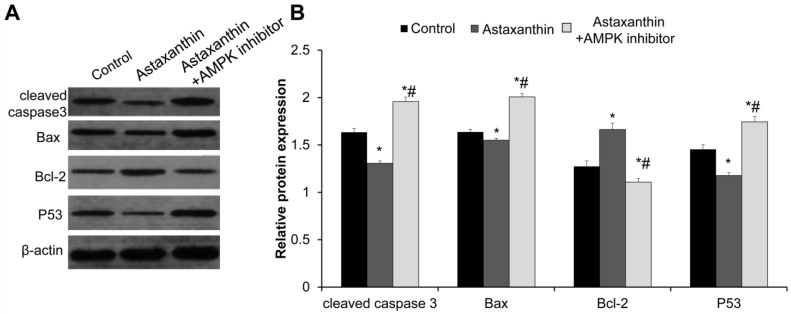
Effects of astaxanthin on expression of apoptosis proteins in follicles. (**A**) Western blot detection of expression of apoptosis proteins cleaved caspase 3, Bax, Bcl-2, and P53 in follicles of day 10. (**B**) Analysis of relative expression levels of these proteins using Image J image analysis software. Experiment was conducted in triplicate (N = 3), and 40 follicles were used in each group. * *p* < 0.05, compared with control group. # *p* < 0.05, compared with astaxanthin group.

**Figure 9 ijms-26-02241-f009:**
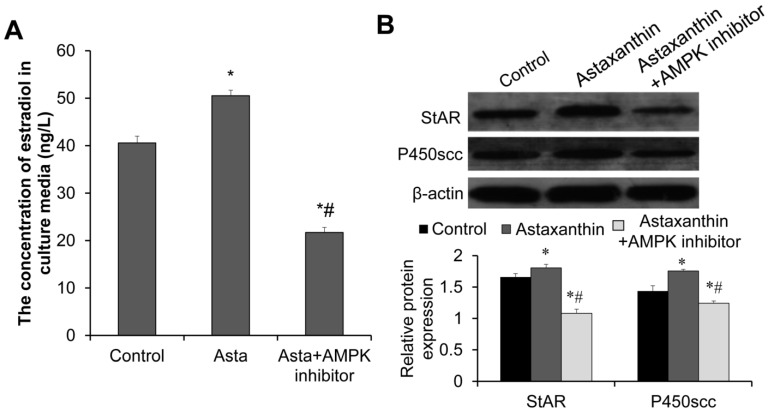
Effects of astaxanthin on estradiol secretion and expression of estrogen synthesis proteins in mouse follicles of day 10. (**A**) Estradiol secretion by follicles of day 10 (ng/L). In each group, 30 follicles were used. (**B**) Western blot analysis of expression of estrogen synthesis proteins StAR and P450scc in follicles of day 10. (The relative expression levels of these proteins were analyzed using Image J image analysis software. In each group, 40 follicles were used. Experiments were repeated three times independently (N = 3). * *p* < 0.05, compared with control group. # *p* < 0.05, compared with astaxanthin group.

**Table 1 ijms-26-02241-t001:** Effects of astaxanthin on development of preantral follicles in vitro (N = 8).

Groups	Number of Oocytes	Survival Rate (%)	Antrum Formation Rate (%)	Maturation Rate (%)
Control	180	178 (98.89 ± 0.96)	153 (85.96 ± 5.15)	121 (79.15 ± 1.84)
DMSO	180	177 (98.33 ± 1.67)	154 (86.99 ± 2.69)	120 (77.96 ± 1.43)
0.25 nM astaxanthin	180	176 (97.77 ± 1.93)	156 (88.64 ± 0.72)	125 (79.62 ± 0.78)
2.5 nM astaxanthin	180	178 (98.89 ± 1.93)	168 (94.38 ± 0.91) *#	143 (85.12 ± 0.80) *#
25 nM astaxanthin	180	175 (97.21 ± 0.96)	128 (73.16 ± 3.30) *#■	64 (49.89 ± 6.00) *#■

Note: * *p* < 0.05, compared with control group; # *p* < 0.05, compared with 0.25 nM astaxanthin group; ■ *p* < 0.05, compared with 2.5 nM astaxanthin group.

**Table 2 ijms-26-02241-t002:** Primer sequences for qRT-PCR analysis.

Gene Name	Primer Sequences (5′ to 3′)		Product Size (bp)	Annealing Temperature (°C)
	Forward	Reverse		
GSH	GAGAGCGTCAACAGGGAGATG	CCAGCCTCCGTTATCCTGGA	249	59
SOD1	ATGCCCATGCTACAGAGGAG	AGACTGGCCCTTCTTGGTCT	143	59
CO1	TCCAACTCATCCCTTGACATCGTGC	TGGCGAAGTGGGCTTTTGCTCA	172	59
CO2	ATTGCCCTCCCCTCTCTACGCATT	CCAGGTTTTAGGTCGTTTGTTGGGA	167	58
CO3	ACTGGAGCCTTTTCAGCCCTCCTT	AGTGTGGTGGCCTTGGTAGGTT	162	58
ATP6	AAGCTCACTCGCCCACTTCCTT	TGTAAGCCGGACTGCTAATGCCA	118	58
ATP8	TCCCACTAGCACCTTCACCA	TGTTGGGGTAATGAATGAGGCAA	103	57
TOM20	AGATGTGGGGCTTTGGCACTGT	AGGTGAGCTGGGGTGCAACATT	198	58
β-actin	TGTTACCAACTGGGACGACA	CTGGGTCATCTTTTCACGGT	146	59

Note: GSH: glutathione; SOD1: superoxide dismutase 1; CO1: cytochrome c oxidase subunit 1; CO2: cytochrome c oxidase subunit 2; CO3: cytochrome c oxidase subunit 3; ATP6: ATP synthase F0 subunit 6; ATP8: ATP synthase F0 subunit 8; TOM20: translocase of the outer membrane subunit 20.

## Data Availability

The raw data supporting the conclusions of this article are available in jianguoyun at https://www.jianguoyun.com/p/DXG7FHsQ2rKbDRiHvPUFIAA (assessed on 25 January 2025) (Password: hL49W5).

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
