# Peer review of "Astaxanthin Alleviates Oxidative Stress in Mouse Preantral Follicles and Enhances Follicular Development Through the AMPK Signaling Pathway"

_ijms, 2025, doi:10.3390/ijms26052241_

Round 1
Reviewer 1 Report
Comments and Suggestions for Authors
Comments to Authors
In this study, authors have investigated effect of astaxanthin on oxidative stress, mitochondrial function, and follicles development mainly focusing on AMPK. Astaxanthin is a strong antioxidant which protect cells to save from inflammation, and have potential skin health, eye health, and cardiovascular health benefits. AMPK-pathway is activated in response to increases in the cellular AMP:ATP ratio when the cell undergoes energetic stress and promotes fatty acid oxidation. This study first time dissect role of astaxanthin in alleviation of oxidative stress by targeting AMPK pathway. Astaxanthin decreases ROS in oocytes and increases expression of mitochondrial biogenesis related proteins, antioxidant proteins, mitophagy related proteins and antiapoptotic proteins. Notably, inhibitor of AMPK reversed effect of astaxanthin. Jiaqi et al reported that astaxanthin enhances in-vitro development of follicles, alleviates oxidative stress in preantral follicles, promotes mitochondrial functioning, mediated by AMPK pathway. The authors have designed and conducted study in very adequate manners, but still there some major concerns which need to be addressed before acceptance for publication.
Comments:
Major Concerns
- "AMPK inhibitor reversed the effects of astaxanthin" is critical but lacks detail. Did it completely negate all effects, or were some pathways still active? Consider specifying which markers were most affected.
- Both mitochondrial biogenesis and mitophagy related proteins are upregulated, interpret these results and elaborate how these antagonistic function related proteins are upregulated under the effect of astaxanthin. This statement is not consistent with reference 47.
- In abstract - mitochondrial genes (CO1, CO2, CO3, ATP6, ATP8, and TOM20) …….. mention clearly what is the role of these genes and why their upregulation is important.
- In abstract - significant increases and decreases in several biomarkers, but no specific percentages or fold changes are provided. Including quantitative data (even in brief) would strengthen the impact of the findings.
- In results - Specify the total number of follicles analyzed per group to improve statistical reliability.
Minor Concerns
- Reference 26 is irrelevant, modify or delete.
- Reference 38 is irrelevant, modify or delete.
- Reference 51 is irrelevant, modify or delete.
- Reference 57 is irrelevant, modify or delete.
Author Response
Major Concerns
- "AMPK inhibitor reversed the effects of astaxanthin" is critical but lacks detail. Did it completely negate all effects, or were some pathways still active? Consider specifying which markers were most affected.
Response: Thanks for the comments. In our study, we found that astaxanthin (2.5 nM) significantly enhanced both the antrum formation (from 85.96% in the control group to 94.38% in the astaxanthin group) and maturation rates (from 79.15% to 85.12%) of oocytes (P<0.05). From Day 4 of in vitro culture, astaxanthin notably increased the area of follicle attachment (from 0.06 µm2 to 0.32 µm2) and the secretion of estradiol (from 32.10 ng/L to 49.73 ng/L) (P<0.05). Additionally, it significantly decreased malondialdehyde content (from 80.54 μM to 62.65 μM) within the follicles while increasing the mRNA expression levels of glutathione and superoxide dismutase 1 (P<0.05). Astaxanthin also reduced reactive oxygen species levels in oocytes (P<0.05). Notably, it enhanced the expression of p-AMPK and PGC-1α, which are key proteins for the AMPK pathway; NRF1 and TFAM, which are crucial for mitochondrial biogenesis; NRF2 and HO-1, which protect against oxidative stress; CO1, CO2, CO3, ATP6, ATP8, and TOM20, which are essential for electron transport chain activity and ATP synthesis; PINK1, Parkin, and LC3-II, which are involved in mitophagy; the Bcl-2, which inhibits apoptosis; and StAR and P450scc, which promotes estrogen synthesis (P<0.05). Furthermore, astaxanthin improved mitochondrial membrane potential and decreased the expression of cleaved caspase 3, Bax, and P53, which promotes cell apoptosis (P<0.05). However, these changes induced by astaxanthin were completely reversed by AMPK inhibitors, indicating the involvement of the AMPK pathway.
To enhance clarity, we have emphasized that all results from astaxanthin treatment were completely reversed by the AMPK inhibitor in our revised manuscript. We have added a detailed description in the Results section and further discussed the implications of these findings in the Discussion section. Please kindly review.
- Both mitochondrial biogenesis and mitophagy related proteins are upregulated, interpret these results and elaborate how these antagonistic function related proteins are upregulated under the effect of astaxanthin. This statement is not consistent with reference 47.
Response: Thanks for the comment. The simultaneous upregulation of mitochondrial biogenesis and mitophagy-related proteins by astaxanthin may initially seem contradictory, as these processes are often considered antagonistic. However, they are complementary mechanisms that work in concert to maintain mitochondrial quality and cellular homeostasis. Mitochondrial biogenesis generates new, healthy mitochondria to replace damaged ones, while mitophagy selectively removes dysfunctional or damaged mitochondria to prevent the accumulation of ROS and maintain a healthy mitochondrial network [1]. Under conditions of oxidative stress, such as those induced by high oxygen levels during in vitro follicle culture, both processes are essential for restoring mitochondrial function and ensuring cellular survival [2, 3]. In our study, the upregulation of astaxanthin on both mitochondrial biogenesis (via p-AMPK, PGC-1α, NRF1, and TFAM) and mitophagy (via PINK1, Parkin, and LC3-II) suggests that it promotes a dynamic balance between mitochondrial renewal and degradation. We suppose that this dual action ensures that damaged mitochondria are efficiently removed while new, functional mitochondria are generated to meet the energy demands of the cell. Such a coordinated response is critical for alleviating oxidative stress and improving follicular development.
To enhance clarity, we have revised the Discussion section to provide a more detailed interpretation of these findings and clarified the relationship between mitochondrial biogenesis and mitophagy in the context of astaxanthin’s effects. Please kindly review.
Additionally, we apologize for the inconsistencies regarding the original reference 47 in our manuscript. In our revised version, we have replaced the original reference with the correct one (the new reference 45) [4].
References:
[1] Liu, L.; Li, Y.; Chen, G.; Chen, Q. Crosstalk between mitochondrial biogenesis and mitophagy to maintain mitochondrial homeostasis. J Biomed Sci 2023, 30(1), 86.
[2] Lewis Luján, L.M.; McCarty, M.F.; Di Nicolantonio, J.J.; Gálvez Ruiz, J.C.; Rosas-Burgos, E.C.; Plascencia-Jatomea, M.; Iloki Assanga, S.B. Nutraceuticals/drugs promoting mitophagy and mitochondrial biogenesis may combat the mitochondrial dysfunction driving progression of dry age-related macular degeneration. Nutrients 2022, 14(9), 1985.
[3] Figueiredo-Pereira, C.; Villarejo-Zori, B.; Cipriano, P.C.; Tavares, D.; Ramírez-Pardo, I.; Boya P.; Vieira, H.L.A. Carbon mon-oxide stimulates both mitophagy and mitochondrial biogenesis to mediate protection against oxidative stress in astrocytes. Mol Neurobiol 2023, 60(2), 851-863.
[4] Chen, Y.; Li, S.; Guo, Y.; Yu, H.; Bao, Y.; Xin, X.; Yang, H.; Ni, X.; Wu, N.; Jia, D. Astaxanthin attenuates hypertensive vascular remodeling by protecting vascular smooth muscle cells from oxidative stress-induced mitochondrial dysfunction. Oxid Med Cell Longev 2020, 2020, 4629189.
- In abstract - mitochondrial genes (CO1, CO2, CO3, ATP6, ATP8, and TOM20) …….. mention clearly what is the role of these genes and why their upregulation is important.
Response: We thank the reviewer for the suggestion to clarify the roles of these genes/proteins. As the abstract has limited space, we have added a brief explanation in the abstract and provided a more detailed description in the Discussion sections of the manuscript.
Please refer to the following information and the revised manuscript for details.
Excerpted abstract
Astaxanthin also reduced reactive oxygen species levels in oocytes (P<0.05). Notably, it enhanced the expression of p-AMPK and PGC-1α, which are key proteins for the AMPK pathway; NRF1 and TFAM, which are crucial for mitochondrial biogenesis; NRF2 and HO-1, which protect against oxidative stress; CO1, CO2, CO3, ATP6, ATP8, and TOM20, which are essential for electron transport chain activity and ATP synthesis; PINK1, Parkin, and LC3-II, which are involved in mitophagy; the Bcl-2, which inhibits cell apoptosis; and StAR and P450scc, which promotes estrogen synthesis (P<0.05). Furthermore, astaxanthin improved mitochondrial membrane potential and decreased the expression of cleaved caspase 3, Bax, and P53, which promotes cell apoptosis (P<0.05). However, these changes induced by astaxanthin were completely reversed by AMPK inhibitor, indicating the involvement of the AMPK pathway.
- In abstract - significant increases and decreases in several biomarkers, but no specific percentages or fold changes are provided. Including quantitative data (even in brief) would strengthen the impact of the findings.
Response: We thank the reviewer for this valuable suggestion. To strengthen the impact of our findings, we have now included specific quantitative data in the abstract and the results section. For example, astaxanthin significantly decreased malondialdehyde content (80.54 μM to 62.65 μM) within the follicles. These quantitative details have been added to provide a clearer representation of the results.
Please refer to the following information and the revised manuscript for details.
Excerpted abstract:
Astaxanthin (2.5 nM) significantly enhanced both the antrum formation (from 85.96% in the control group to 94.38% in the astaxanthin group) and maturation rates (from 79.15% to 85.12%) of oocytes (P<0.05). From Day 4 of in vitro culture, astaxanthin notably increased the area of follicle attachment (from 0.06 µm2 to 0.32 µm2) and the secretion of estradiol (from 32.10 ng/L to 49.73 ng/L) (P<0.05). Additionally, it significantly decreased malondialdehyde content (from 80.54 μM to 62.65 μM) within the follicles while in-creasing the mRNA expression levels of glutathione and superoxide dismutase 1 (P<0.05).
- In results - Specify the total number of follicles analyzed per group to improve statistical reliability.
Response: We agree with the reviewer that specifying the number of follicles analyzed is important for statistical reliability. In the revised manuscript, we have added the total number of follicles analyzed per group. For example, in Figure 3C, 30 follicles per group were analyzed. This information has been included in the Results section to enhance the transparency and reliability of our statistical analysis. Please kindly review.
Minor Concerns
- Reference 26 is irrelevant, modify or delete.
- Reference 38 is irrelevant, modify or delete.
- Reference 51 is irrelevant, modify or delete.
- Reference 57 is irrelevant, modify or delete.
Response: We sincerely apologize for the inclusion of irrelevant references. Upon careful review, we have deleted the references 26, 38, 51, and 57, as they were not pertinent to the context in which they were cited. We have also ensured that all remaining references are relevant and appropriately support the content of the manuscript.
Reviewer 2 Report
Comments and Suggestions for Authors
Dear authors,
Thank you for your interesting article about astaxanthin, a carotenoid colour and antioxidant.
Searching for antioxidants, which can improve follicles development or alleviates the negative surrounding effects during in vitro cultivation of follicles, is important nowadays, because also humans have a big problem with fertility. So, I think that any substance /chemical which can help is needed.
Introduction is written well, methods are described step by step, so they are reusable.
In results section the obtained results are described very well, it is easy understandable, and figures with graphs are nice, statistical significance is clear. Discussion gives us additional info about previously obtained results and mechanisms of astaxanthin function.
I have some comments and questions.
- Usually the phrase “in vitro” is written in italics: in vitro. Please check it in whole document.
- Part Methods: I did not find the info about the solvent of astaxanthin. Probably it is DMSO, so please add this info in the text.
- Part result: I did not find the test figures number 8 a 9, they are missing. Please add them. There is only title of Figures.
- Question: Astaxanthin is freely used as a nutritional supplement, and you can buy it in pharmacy. Do you have any info if it is used in human artificial fertilization and processes connected with it (e.g. collection of human oocytes for fertilization)? If yes, please add this info into the text.
Author Response
- Usually the phrase “in vitro” is written in italics: in vitro. Please check it in whole document.
Response: As suggested, we have written the phrase “in vitro” is written in italics: in vitro. Please refer to the revised manuscript for details.
- Part Methods: I did not find the info about the solvent of astaxanthin. Probably it is DMSO, so please add this info in the text.
Response: In our study, the astaxanthin was dissolved in DMSO. As suggested, we have added this to the Methods section. Please kindly review.
- Part result: I did not find the test figures number 8 a 9, they are missing. Please add them. There is only title of Figures.
Response: Figures 8 and 9 may have been missed during the submission. We apologize for any confusion caused. Now, we have provided these figures. Please kindly review.
- Question: Astaxanthin is freely used as a nutritional supplement, and you can buy it in pharmacy. Do you have any info if it is used in human artificial fertilization and processes connected with it (e.g. collection of human oocytes for fertilization)? If yes, please add this info into the text.
Response: The reviewer raised a very good point. Based on recent studies, astaxanthin has indeed been used in clinical settings to improve reproductive outcomes in women undergoing assisted reproductive technology. We have added this information to the Discussion section of the manuscript to highlight the translational relevance of our findings.
Please refer to the following information and the revised manuscript for details.
Astaxanthin has shown promising applications in human-assisted reproductive technology. For instance, in women with endometriosis-related infertility, astaxanthin supplementation (6 mg/day for 12 weeks) significantly increased the number of retrieved oocytes and improved embryo quality [1]. Similarly, in patients with polycystic ovary syndrome, astaxanthin treatment (12 mg/day for 60 days or 8 mg/day for 40 days) enhanced the rates of high-quality oocytes and embryos, as well as oocyte maturation rates [2-4]. Furthermore, in poor ovarian responders, astaxanthin supplementation (12 mg/day for 8 weeks) improved the number of retrieved oocytes, mature oocytes, frozen embryos, and high-quality embryos [5]. These studies demonstrate that astaxanthin plays a significant role in improving oocyte and embryo quality in humans. While astaxanthin has been widely used in the in vitro culture of oocytes from mice, pigs, and cattle [6-9], its application in the in vitro culture of human oocytes or follicles remains unexplored, likely due to the challenges associated with obtaining human oocytes and the immaturity of in vitro maturation systems. Nonetheless, these findings underscore the potential of astaxanthin as a therapeutic agent in human-assisted reproductive technology and provide a strong rationale for further investigation into its mechanisms and applications.
References:
[1] Rostami S, Alyasin A, Saedi M, Nekoonam S, Khodarahmian M, Moeini A, Amidi F. Astaxanthin ameliorates inflammation, oxidative stress, and reproductive outcomes in endometriosis patients undergoing assisted reproduction: A randomized, triple-blind placebo-controlled clinical trial. Front Endocrinol (Lausanne), 2023, 14: 1144323.
[2] Jabarpour M, Aleyasin A, Nashtaei MS, Lotfi S, Amidi F. Astaxanthin treatment ameliorates ER stress in polycystic ovary syndrome patients: a randomized clinical trial. Sci Rep, 2023, 13(1): 3376.
[3] Gharaei R, Alyasin A, Mahdavinezhad F, Samadian E, Ashrafnezhad Z, Amidi F. Randomized controlled trial of astaxanthin impacts on antioxidant status and assisted reproductive technology outcomes in women with polycystic ovarian syndrome. J Assist Reprod Genet, 2022, 39(4): 995-1008.
[4] Fu X, Cao W, Ye F, Bei J, Du Y, Wang L. Astaxanthin compound nutrient improved insulin resistance, hormone levels, embryo quality and pregnancy outcomes in polycystic ovary syndrome patients undergoing in vitro fertilization/intracytoplasmic sperm injection. Drug Discov Ther, 2024, 18(5): 296-302.
[5] Shafie A, Aleyasin A, Saffari M, Saedi M, Rostami S, Rezayi S, Mohammadi SD, Amidi F. Astaxanthin improves assisted reproductive technology outcomes in poor ovarian responders through alleviating oxidative stress, inflammation, and apoptosis: a randomized clinical trial. J Ovarian Res, 2024, 17(1): 212.
[6] Tana C, Somsak P, Piromlertamorn W, Sanmee U. Effects of astaxanthin supplementation in fertilization medium and/or culture medium on the fertilization and development of mouse oocytes. Clin Exp Reprod Med, 2022, 49(1): 26-32.
[7] Li Y, Dong Z, Liu S, Gao F, Zhang J, Peng Z, Wang L, Pan X. Astaxanthin improves the development of the follicles and oocytes through alleviating oxidative stress induced by BPA in cultured follicles. Sci Rep, 2022, 12(1): 7853.
[8] Yang X, Zhou D, Gao L, Wang Y, Wang Y, Jia R, Bai Y, Shi D, Lu F. Effects of Astaxanthin on the Physiological State of Porcine Ovarian Granulose Cells Cultured In Vitro. Antioxidants (Basel), 2024, 13(10): 1185.
[9] Abdel-Ghani MA, Yanagawa Y, Balboula AZ, Sakaguchi K, Kanno C, Katagiri S, Takahashi M, Nagano M. Astaxanthin improves the developmental competence of in vitro-grown oocytes and modifies the steroidogenesis of granulosa cells derived from bovine early antral follicles. Reprod Fertil Dev, 2019, 31(2): 272-281.
Round 2
Reviewer 1 Report
Comments and Suggestions for Authors
Authors have addresses all comments properly and improved manuscript.